# Rivalry between pitch and timbre in auditory stream segregation

**Geng-Yan Jhang**[1]*, **Kazuo Ueda**[2,3]*, **Hiroshige Takeichi**[4], **Gerard B. Remijn**[2], **Emi Hasuo**[2]

**1** Human Science International Course, Graduate School of Design, Kyushu University, Fukuoka, Japan, **2** Department of Acoustic Design, Faculty of Design/Research Center for Applied Perceptual Science, Kyushu University, Fukuoka, Japan, **3** Research and Development Center for Five-Sense Devices, Kyushu University, Fukuoka, Japan, **4** Open Systems Information Science Team, Advanced Data Science Project (ADSP), RIKEN Information R&D and Strategy Headquarters (R-IH), RIKEN, Yokohama, Kanagawa, Japan

* jhang.gengyan@gmail.com (G-YJ); ueda@design.kyushu-u.ac.jp; ueda.psychoacoustics@kyudai.jp (KU)

**Data availability statement:** All relevant data are within the paper and its Supporting information files.

## Abstract

Two rapidly alternating tones with different pitches may be perceived as one integrated stream when the pitch differences are small or two separated streams when the pitch differences are large. Likewise, timbre differences between two tones may also cause such *sequential stream segregation*. Moreover, the effects of pitch and timbre on stream segregation may cancel each other out, which is called a trade-off. However, how timbre differences caused by specific patterns of spectral shapes interact with pitch differences and affect stream segregation has been largely unexplored. Therefore, we used *stripe tones*, in which stripe-like spectral patterns of harmonic complex tones were realized by grouping harmonic components into several bands based on harmonic numbers and removing harmonic components in every other band. Here, we show that 2- and 4-band stimuli elicited distinctive stream segregation against pitch proximity. By contrast, pitch separations dominated stream segregation for 16-band stimuli. The results for 8-band stimuli most clearly showed the trade-off between pitch and timbre on stream segregation. These results suggest that the stimuli with a small number (≤4) of bands elicit strong stream segregation due to sharp timbral contrasts between stripe-like spectral patterns, and that the auditory system looks to be limited in integrating blocks of frequency components dispersed over frequency and time.

## Introduction

*Auditory scene analysis* [1] refers to the auditory function in which we perceptually organize the outer world filled with a mixture of sounds. Sometimes, we can exploit binaural cues to separate several sounds, but the cues are not always available or practical. Even in such situations, normal-hearing listeners usually successfully pick up a target from the background sounds, like a talker's voice talking on a noisy railway platform over a phone. Linguistic cues

**Funding:** This research was supported by the Japan Society for the Promotion of Science (JSPS; https://www.jsps.go.jp) KAKENHI Grant No. JP19H00630 for Kazuo Ueda, and by the Japan-Taiwan Exchange Association (https://www.koryu.or.jp) with a scholarship for Geng-Yan Jhang under the supervision of Kazuo Ueda. The funders had no role in study design, data collection and analysis, decision to publish, or preparation of the manuscript. There was no additional external funding received for this study.

should contribute a lot to segregating speech and background noise in such a situation; nevertheless, the perceptual characteristics of the auditory system should be involved in the segregation process. Furthermore, if both a target and background are nonspeech, like a warning signal in background noise, we must make use of other cues, such as pitch and timbre. The auditory system can organize and segregate a sound mixture into more than two streams with the pitch and timbre differences of the sounds. The current investigation focuses on how pitch and timbre interact in *auditory stream segregation* [2].

Miller and Heise [3] reported that two rapidly alternating pure tones (i.e., tones having only one frequency component and a sinusoidal waveform) with some slight frequency differences are heard as a trill, a continuously changing pitch pattern. However, the two tones with a wide frequency separation are perceived as two separate and unrelated melodies. Since then, *sequential stream segregation* has been a widely studied research topic [4–22]. In particular, low-high-low (LHL_) and high-low-high (HLH_) tone patterns, i.e., so-called *triplets* (Fig 1), are the standard research tools to investigate sequential stream segregation caused by pitch differences. Here, "L" signifies a "low-pitch tone," "H" a "high-pitch tone," and "_" a "silent gap". In this connection, we define a "(fundamental) frequency separation" as a "(fundamental) frequency difference between a low-pitch tone and a high-pitch tone, expressed in semitone units" in this article.

Suppose a series of "LHL_" triplets is presented. If the tone sequence is perceptually grouped, then a listener will hear one stream with a galloping rhythm (LHL_LHL_LHL_...). In contrast, if the tone sequence is perceptually segregated into two streams, a listener will hear a low-pitch-tone sequence with a short gap in between (L_L_L_...) and a high-pitch-tone sequence with a long gap in between (H___H___H___...).

One possibility to increasing probability of segregation is to widen the frequency separation between Ls and Hs [4], and the other possibility is to prolong a sequence to *build up* segregation [5,6]. With an induction sequence prior to a test sequence consisting of three sets of LHL_ tone triplets, Haywood and Roberts [13–15] reported a build-up in sequential stream segregation in participants' responses to the final triplets. At the same time, with only three sets of triplets, they found that a separation of 14 semitones between L and H tones caused segregation around 75%–90% of the time, while a separation of four semitones caused almost

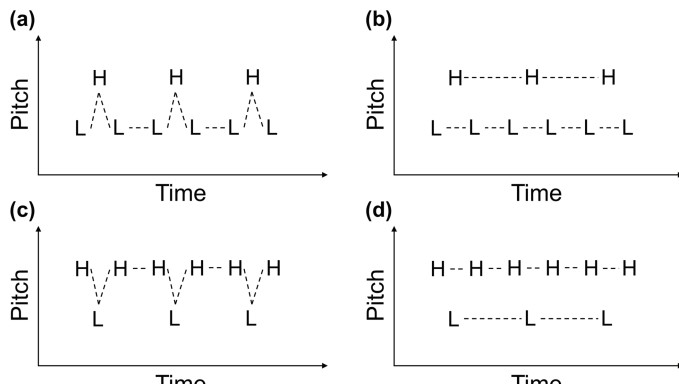

**Fig 1. Schematic illustrations of one-stream (integration) perception and two-stream (segregation) perception.** The sequence in each panel consists of low-pitch and high-pitch tones (Ls and Hs). The dashed lines represent perceptual connections between tones. (a) One-stream perception and (b) two-stream perception for an LHL_ tone pattern ("_" signifies a "silent gap"). (c) and (d) show similar examples for an HLH_ tone pattern.

no segregation. With a 10-semitone separation, the percentages of segregation responses came in between (around 25%–55%).

Harmonic complex tones have been used in some studies on sequential stream segregation. A harmonic complex tone consists of a series of pure tones called harmonics with frequencies that are integer multiples of a fundamental frequency, i.e., the lowest frequency. The pitch of a harmonic complex tone usually corresponds to the pitch of the fundamental; however, even if the fundamental frequency component is missing, the pitch of the harmonic complex tone does not change. This is called the missing fundamental phenomenon. When one modifies the level balance of frequency components in a harmonic complex tone, timbre varies, although this is not the only factor that affects the timbre of a harmonic complex tone.

Fundamental frequency separation is a prominent cue for segregation when harmonic complex tones have clear pitches with resolvable harmonics, that is, usually the first five to eight harmonics [23,24]. In addition, even without resolvable harmonics, fundamental frequency separation between bandpass-filtered harmonic complex tones induces stream segregation [23,25–27]. A sequence of amplitude-modulated noises may be perceptually segregated based on pitch separation [28–30].

Other factors related to timbre can influence stream segregation: differences in spectral regions [4,7,27,31], the number of harmonics [9], the spectral composition between odd- and even-numbered harmonics [8], spectral peaks [32], spectral slopes [33–35], and the amplitude envelope patterns across harmonics, i.e., spectrum variation over time [12].

Some studies have focused on the combined effects of pitch and timbre differences on sequential stream segregation. Pitch and timbre have been revealed to be competitive [7,32] or interactive [34] in stream segregation. Thus, the effects of pitch and timbre on stream segregation may cancel each other out. We call this relationship hereafter a *trade-off* between pitch and timbre in stream segregation. In these studies, timbre was manipulated by shifting a set of four consecutive harmonics [7], moving a spectral peak [32], or tilting the overall spectral slope [34].

However, these studies left room open to investigate how timbre differences caused by specific patterns of spectral shapes affect stream segregation. The most recent investigations on degraded speech perception provide clues for extending previous investigations on sequential stream segregation. It has been established that speech intelligibility drastically varies when speech sentences are interrupted in time and frequency [36,37]. Ueda et al. call such speech *checkerboard speech* because its spectrogram looks like a checkerboard. The intelligibility of checkerboard speech varies drastically depending on the combination of the number of frequency bands and segment duration. The intelligibility of 20- or 16-band checkerboard speech is almost at the ceiling, irrespective of segment duration. However, the intelligibility of two- or four-band checkerboard speech is highest (more than 90%) at 20-ms segment duration, lowest (around 35%–40%) at around 80–160 ms, and moderate (more than 50%) at 320 ms. We may conjecture that spectrotemporal interruption with two or four frequency bands and 80–160-ms segment duration promotes auditory segregation, hampers integration, and reduces intelligibility.

To test this hypothesis, it is necessary to examine how the auditory system integrates or segregates nonspeech stimuli, rapidly switching in time and frequency. Among variables that can be included, we selected fundamental frequency shifts and harmonic number band switching. The fundamental frequency shifts result in pitch shifts, and harmonic number band switching varies timbre. To realize drastic changes in timbre, we group harmonic components into several bands based on harmonic numbers and remove harmonics in every other band. We call these stimuli *stripe tones*. Furthermore, two extreme combinations of pitch and timbre shift directions—a *congruent* shift and an *incongruent* shift—are constructed to capture the

whole picture of the possible combinations of these variables. The congruent shift is defined as the combination of upward fundamental frequency shifts and an odd-numbered-band tone to an even-numbered-band tone shift and vice versa, and the incongruent shift is defined as the combination of upward fundamental frequency shifts and an even-numbered-band tone to an odd-numbered-band tone shift and vice versa.

We constructed *stripe tone sequences* with the LHL_ and HLH_ triplet patterns. LHL_ tone sequences were combined with congruent shifts in experiment 1 and incongruent shifts in experiment 2 (Fig 2). HLH_ tone sequences were combined with congruent shifts in experiment 3 and incongruent shifts in experiment 4 (Fig 3).

The following three research questions were raised.

1. Do timbre differences in stripe tones affect the percentages of segregation responses?
2. Does congruency in shift directions of pitch and timbre for stripe tone sequences affect percentages of segregation responses?
3. Do the stripe tone sequence patterns, LHL_ and HLH_, interact with the congruency of pitch and timbre shift directions?

With these research questions, we planned to clarify a trade-off between pitch and timbre in sequential stream segregation with stimuli characterized by stripe-like spectral patterns. We found timbre manipulated by the number of bands dominated the results when the number of bands was small (cf. Jhang et al. [38]).

## Method

### Participants

Eighteen paid listeners (age range: 20–28; mean age: 22.7) with normal hearing (audiometric thresholds ≤ 25 dB HL at every octave point from 250 to 8000 Hz, screened with an audiometer, Rion AA-58, Rion Corp., Kokubunji, Japan), participated in the series of experiments. Participants were recruited from Japanese-native students on the Ohashi campus of Kyushu University. The recruitment period for experiments 1 and 2 was from June 15 to July 20, 2022, and for experiments 3 and 4 from September 12 to October 8, 2022. Absolute pitch possessors were screened out based on self-reports. All participants but one had extracurricular musical training (10.1 years on average). This research was conducted with prior approval of the Ethics Committee of Kyushu University (approval ID: 72). All the participants signed a written informed consent.

### Conditions

Both pure tones and (full-band) harmonic complex tones with 35 consecutive harmonics were used as control stimuli (Fig 4) as well as exemplars (Fig 5). Stripe tones were used as experimental stimuli.

Exemplars' (fundamental) frequency separations were 2 semitones for "one-stream" exemplars and 18 semitones for "two-stream" exemplars. Three variables were manipulated in the control conditions: (fundamental) frequency separation (4, 10, and 16 semitones), control stimulus type (pure tone and full-band harmonic complex tone), and tone sequence pattern (LHL_ and HLH_). The range of (fundamental) frequency separations was determined referring to Haywood and Roberts [13–15] and the results of our pilot experiment.

Four variables were manipulated in the stripe-tone conditions: fundamental frequency separation (4, 10, and 16 semitones), number of bands (2, 4, 8, and 16 bands in Table 1), congruency (congruent and incongruent), and tone sequence pattern (LHL_ and HLH_).

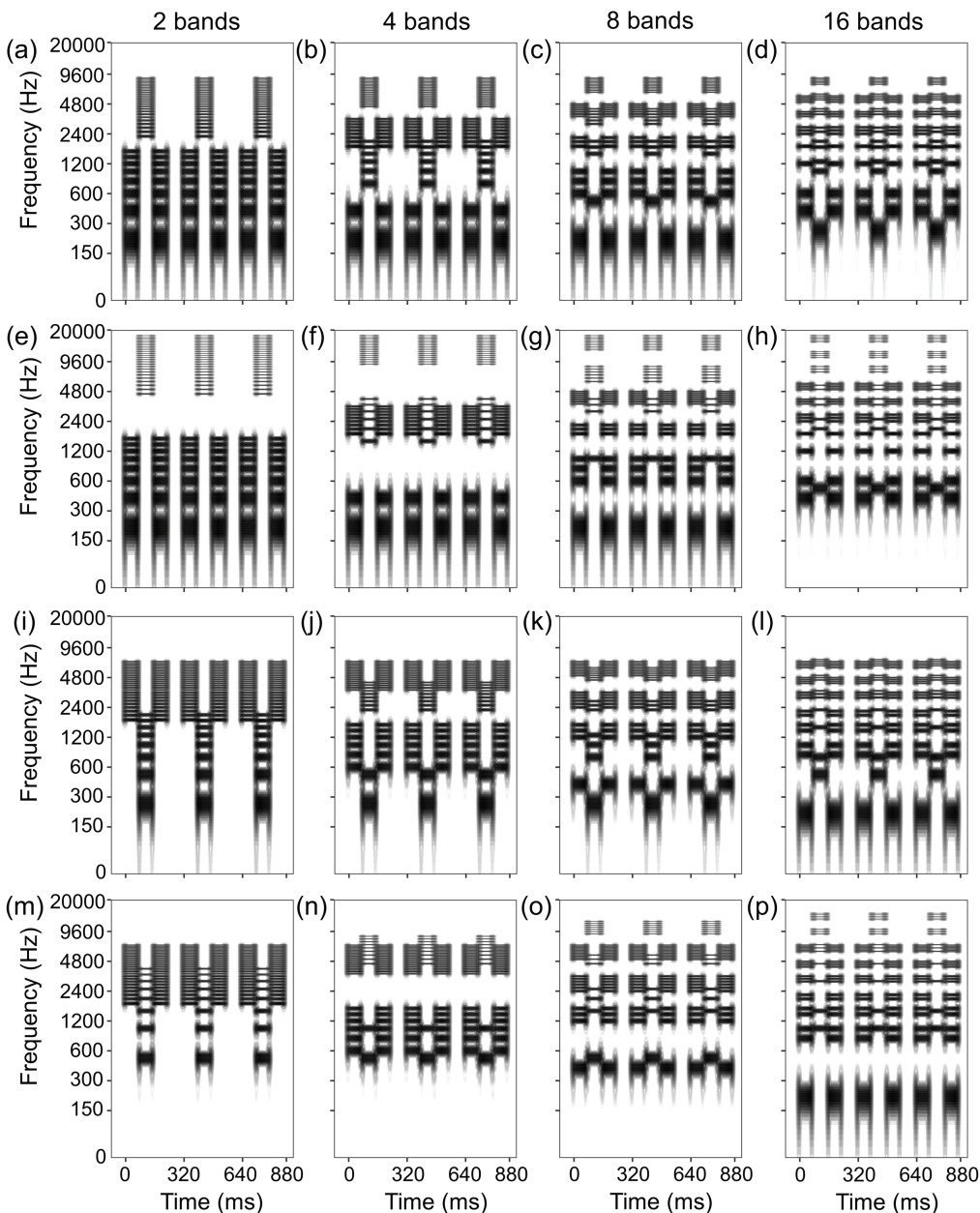

**Fig 2. Spectrograms of stripe tones with LHL_ triplets.** Stripe tones are harmonic complex tones divided into 2, 4, 8, or 16 bands. Frequency components are grouped into several bands based on harmonic numbers (Tables 1 and 2), and the components in every other band are removed. The tone duration is 80 ms. Two possible spectral patterns (with only odd- or even-numbered bands) are switched between L and H tones to make a sequence. An LHL_ ("_" denotes a "silent gap") triplet is presented three times with 80-ms gaps. The columns are in the order of 2-, 4-, 8-, and 16-band stimuli from left to right. The upper two rows (a–h) represent congruent sequences used in experiment 1 (fundamental frequencies and spectral patterns move in the same direction), and the lower two rows (i–p) represent incongruent sequences used in experiment 2 (fundamental frequencies and spectral patterns move in the opposite direction). The first and third rows from the top (a–d and i–l) represent stimuli with a four-semitone fundamental frequency separation. The second and fourth rows (e–h and m–p) represent stimuli with a 16-semitone separation. The L-tone fundamental frequency is fixed at 200 Hz. The H-tone fundamental frequency for the four-semitone separation is 252 Hz, and the 16-semitone separation is 504 Hz.

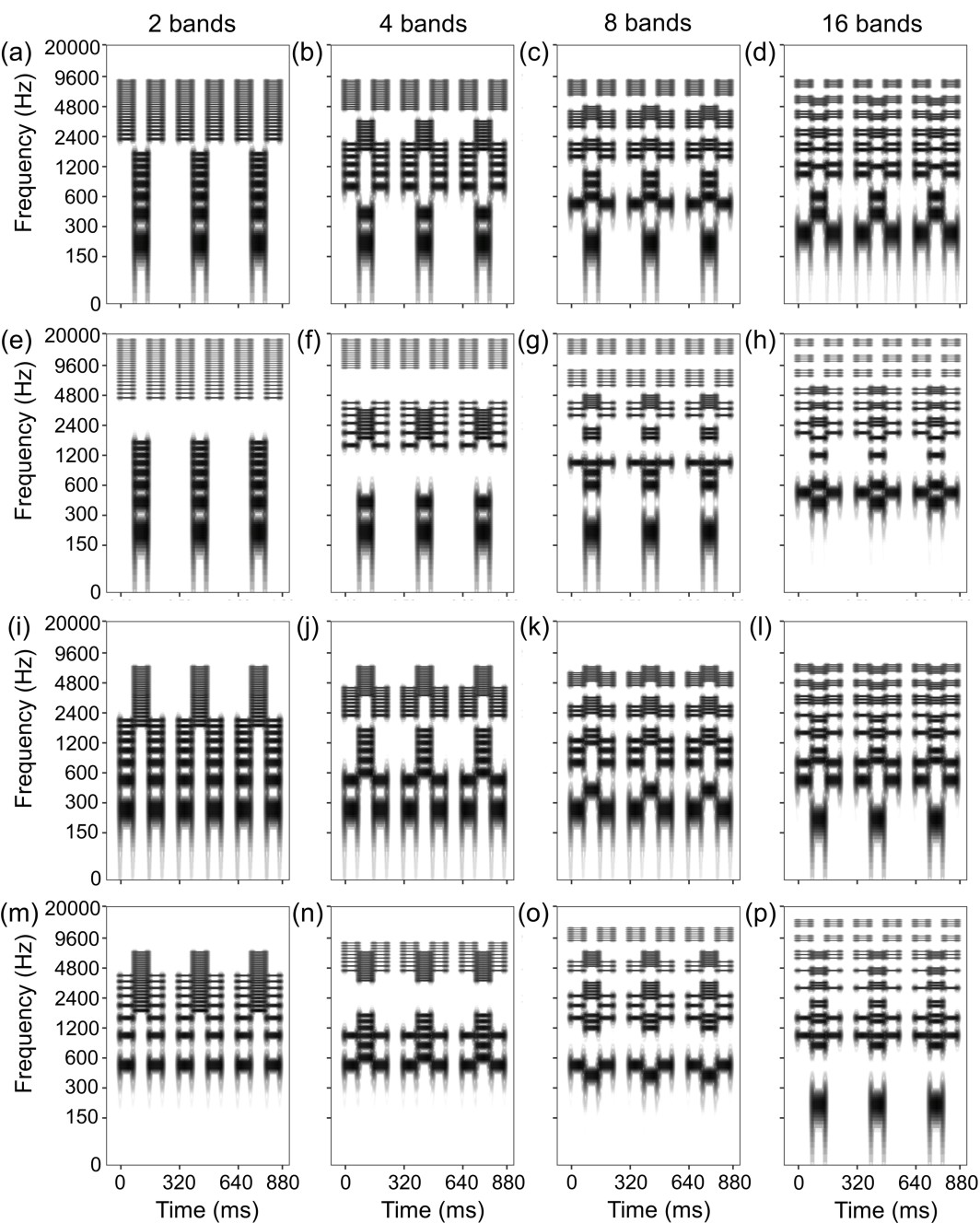

**Fig 3. Spectrograms of stripe tones with HLH_ triplets.** Congruent and incongruent sequences (a–h and i–p) were used in experiments 3 and 4. The columns are in the order of 2-, 4-, 8-, and 16-band stimuli from left to right. The first and third rows (a–d and i–l) represent stimuli with a four-semitone separation, and the second and fourth rows (e–h and m–p) represent stimuli with a 16-semitone separation.

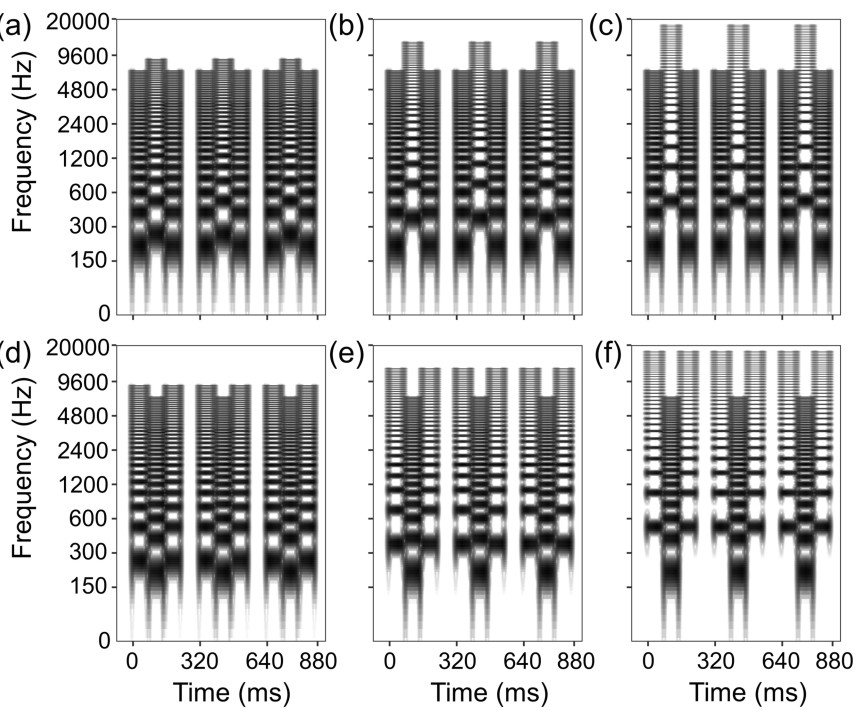

**Fig 4. Spectrograms of control stimuli with (full-band) harmonic complex tones.** LHL_ sequences with (a) 4-, (b) 10-, and (c) 16-semitone separation were used in both experiments 1 and 2. HLH_ sequences with (d) 4-, (e) 10-, and (f) 16-semitone separation were used in both experiments 3 and 4. Pure-tone stimuli were also used but are not shown here.

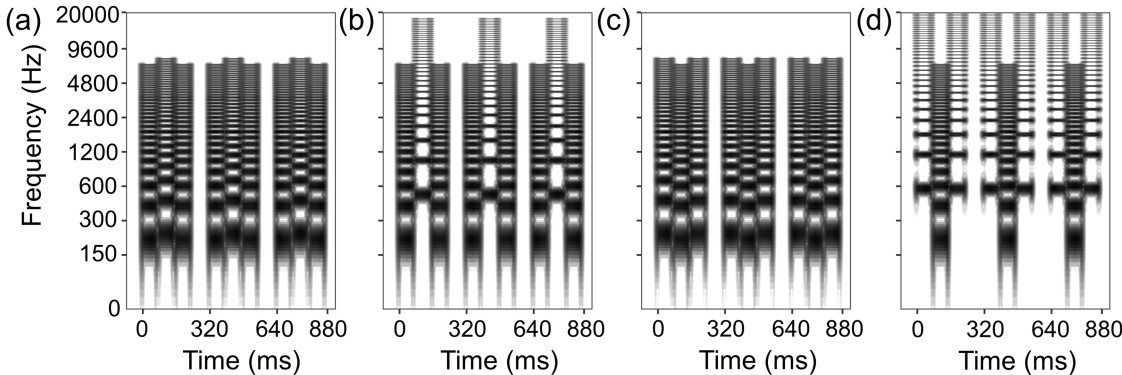

**Fig 5. Spectrograms of audio exemplars with (full-band) harmonic complex tones.** (a) The LHL_ audio exemplar for one stream (integration) with 2-semitone separation and (b) the LHL_ audio exemplar for two streams (segregation) with 18-semitone separation were used in both experiments 1 and 2. HLH_ exemplars for one stream and two streams (c and d) with 2- and 18-semitone separation were used in both experiments 3 and 4. Note that the 2-semitone separation, as in Fig 5(a and c), was narrower than the 4-semitone separation, which was the narrowest (fundamental) frequency separation used in measuring segregation. Likewise, the 18-semitone separation, as in Fig 5(b and d), was wider than the 16-semitone separation, which was the widest (fundamental) frequency separation used in measuring segregation. Pure-tone audio exemplars were also used (not shown here).

**Table 1. The passbands in Hz for L-stripe-tones.**

| Band No. | 2 bands | 4 bands | 8 bands | 16 bands |
|---|---|---|---|---|
| 1 | 50–1600 | 50–570 | 50–300 | 50–175 |
| 2 | 1600–7000 | 570–1600 | 300–570 | 175–300 |
| 3 | | 1600–3400 | 570–1000 | 300–425 |
| 4 | | 3400–7000 | 1000–1600 | 425–570 |
| 5 | | | 1600–2320 | 570–770 |
| 6 | | | 2320–3400 | 770–1000 |
| 7 | | | 3400–4800 | 1000–1270 |
| 8 | | | 4800–7000 | 1270–1600 |
| 9 | | | | 1600–1925 |
| 10 | | | | 1925–2320 |
| 11 | | | | 2320–2800 |
| 12 | | | | 2800–3400 |
| 13 | | | | 3400–4000 |
| 14 | | | | 4000–4800 |
| 15 | | | | 4800–5800 |
| 16 | | | | 5800–7000 |

The 2–16 bands were defined by Ueda et al. [37] based on Ueda and Nakajima [39] and Zwicker and Terhardt [40].

**Table 2. Correspondence between band numbers and harmonic numbers for stripe tones.**

| Band No. | 2 bands | 4 bands | 8 bands | 16 bands |
|---|---|---|---|---|
| 1 | 1–8 | 1–2 | 1 | N.A. |
| 2 | 9–35 | 3–8 | 2 | 1 |
| 3 | | 9–17 | 3–5 | 2 |
| 4 | | 18–35 | 6–8 | N.A. |
| 5 | | | 9–11 | 3 |
| 6 | | | 12–17 | 4–5 |
| 7 | | | 18–24 | 6 |
| 8 | | | 25–35 | 7–8 |
| 9 | | | | 9 |
| 10 | | | | 10–11 |
| 11 | | | | 12–14 |
| 12 | | | | 15–17 |
| 13 | | | | 18–20 |
| 14 | | | | 21–24 |
| 15 | | | | 25–29 |
| 16 | | | | 30–35 |

The correspondence was determined with an L tone (fundamental frequency of 200 Hz) by referring to the passbands in Table 1 and was applied to H tones.

Table 2 shows the correspondence between band numbers and harmonic numbers for stripe tones. Congruency refers to whether or not the direction of fundamental frequency shifts coincides with the direction of spectral pattern movements: *congruent* as in Figs 2(a)–2(h) and 3(a)–3(h) or *incongruent* as in Figs 2(i)–2(p) and 3(i)–3(p). For the full set of stimulus spectrograms, see S1–S6 Figs. For each experiment, six control conditions [three steps of (fundamental) frequency separations × two control stimulus types] and 12 stripe-tone conditions (three steps of fundamental frequency separations × four steps of the number of bands) were constructed. Thus, 18 conditions were prepared for each experiment.

## Stimuli

All stimuli were generated with custom software written in J language [41], with a sampling frequency of 44100 Hz and a quantization of 16 bits. Harmonic complex tone stimuli were generated by adding equal-amplitude components in sine phase. L (fundamental) frequency was fixed at 200 Hz. H (fundamental) frequencies for 2-, 4-, 10-, 16-, and 18-semitone separations were 225, 252, 356, 504, and 566 Hz. The stimulus and gap duration was 80 ms, including 10-ms rise and fall time with raised-cosine amplitude envelopes for a stimulus.

The root-mean-square (RMS) amplitude of each stimulus was equalized. A pure tone of 1000 Hz with the same RMS amplitude was used as a calibration tone. The sound pressure level (SPL) of the calibration tone was adjusted to 70 dB (A). The SPLs at headphone outputs were measured with an artificial ear (Brüel & Kjær type 4153, Brüel & Kjær Sound & Vibration Measurement A/S, Nærum, Denmark), a condenser microphone (Brüel & Kjær type 4192), and a sound level meter (Brüel & Kjær type 2260). Each stimulus SPL was confirmed with a 60-second tone of the same frequency composition.

## Procedure

The experiment was conducted in a double-walled sound-attenuating booth (Music Cabin SD3, Takahashi Kensetsu, Kawasaki, Japan). The stimuli were diotically presented to participants through headphones (Beyerdynamic DT 990 PRO, Beyerdynamic GmbH, Heilbronn, Germany). Custom software written with the LiveCode package [42] was used to present the stimuli and to record participants' responses. The headphones were driven with a universal serial bus (USB) interface (Roland Rubix24, Roland Corp., Shizuoka, Japan) and an amplifier (Luxman, L-505f, Luxman Corp., Yokohama, Japan).

The experimenter explained the concepts of one stream and two streams to the participants using schematic illustrations (Fig 6), written explanations, and audio exemplars. First, participants were instructed that they would hear exemplars of a clearly integrated sequence called "one stream." Then, two-semitone exemplars with pure and full-band tones were presented. Subsequently, they were told that they would hear exemplars sound like a distinctly segregated sequence called "two streams," and 18-semitone exemplars were presented. Participants could ask for presentations to be repeated.

The participants were instructed to focus on the final triplet of the three-triplet sequences and to report whether they heard one stream or two streams (Fig 6). They were also instructed to avoid trying to listen specifically for either integration or segregation, but rather simply report which of the two percepts was more dominant. Participants were instructed to select a "One" or "Two" response button on a computer screen. Their selections were confirmed with a message box each time, providing an opportunity to correct an erroneous input.

Each trial started after a three-second pause to eliminate the build-up effect on stream segregation from previous trials. Each trial block consisted of 18 trials of conditions that were randomly ordered. After two practice trial blocks, 20 main trial blocks were run. Thus, each condition was measured 20 times. Participants were allowed to take a break between the blocks.

A group of nine participants, including the participant who had no extracurricular musical training, were assigned to the experiments in the order of 1, 2, 4, and 3. Another group of nine participants were assigned to the experiments in the order of 2, 1, 3, and 4; however, one participant did not participate in experiments 3 and 4. Therefore, the data from the last mentioned participant was excluded from the following analysis.

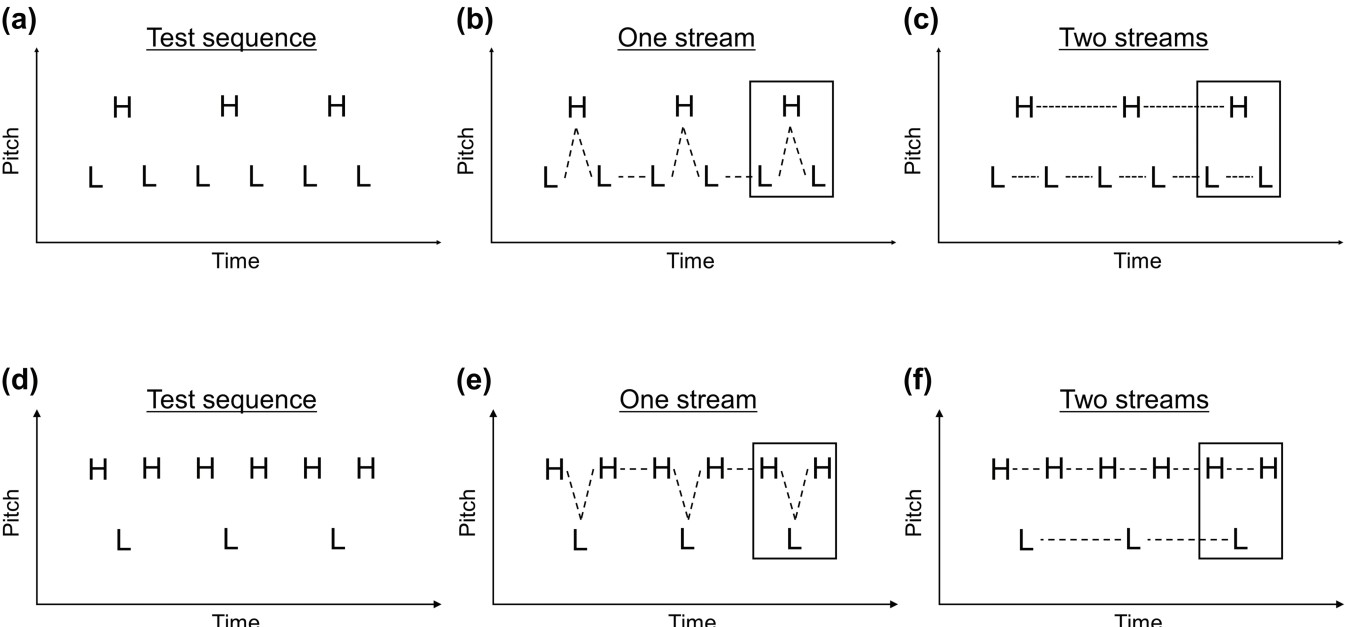

**Fig 6. Schematic illustrations given to participants.** Schematic illustrations for an LHL_ sequence were used in experiments 1 and 2: (a) a test sequence, (b) perception of one stream, and (c) perception of two streams. Similar schematic illustrations were used for an HLH_ sequence in experiments 3 and 4 (d–f). The dashed lines represent perceptual connections between tones. Participants were instructed to focus on the final triplet (marked with squares) and report whether they heard one stream or two streams. Reproduced with translations from Japanese to English.

The series of experiments was conducted on different days for each participant. Each participant completed four experiments within 110 days, taking 60 to 100 minutes to complete each one. Experiments 1 and 2 were conducted within 15 days. Then, after more than 44 days, experiments 3 and 4 were conducted within 17 days.

## Statistical method

Statistical analysis with a generalized linear mixed model (GLMM) was performed with a logit link function as implemented in JMP Pro [43]. The results of control and stripe-tone conditions were separately analyzed. Regarding the control conditions, the data were analyzed for the fixed effects of (fundamental) frequency separation, control stimulus type, tone sequence pattern (all categorical predictors), and their interactions. The analysis model with these fixed effects was fitted to the data with the following candidate sets of random effects: (1) congruency and participant, (2) congruency and extracurricular musical training nested under participant, (3) congruency and trial block (i.e., block number 1–20 in each experiment) nested under participant, and (4) congruency, trial block nested under participant, and experiment order.

Regarding the stripe-tone conditions, the data were analyzed for the fixed effects of fundamental frequency separation, number of bands, congruency, tone sequence pattern (all categorical predictors), and their interactions. The analysis model with these fixed effects was fitted to the data with the following candidate sets of random effects: (1) participant, (2) extracurricular musical training nested under participant, (3) trial block nested under participant, and (4) trial block nested under participant and experiment order. An appropriate

model was selected by examining twice the negative of the residual log pseudo-likelihood ($-2ResidualLogPseudo-Likelihood$) and the ratio of the generalized $\chi^2$ statistic and its degrees of freedom (generalized $\chi^2/df$). The statistical power of the fixed effects was estimated with JMP Pro [43] based on 1000-times simulation at an alpha level of 0.05. Post-hoc multiple comparisons with Tukey's honestly significant difference (HSD) tests were conducted.

## Results

Fig 7 shows the results of experiments 1–4.

### Control conditions

The percentages of segregation responses for pure and full-band harmonic complex tones increased with widening (fundamental) frequency separations. Full-band harmonic complex tones showed higher percentages of segregation responses than pure tones except for the four-semitone separation. The effect of tone-sequence patterns was negligible.

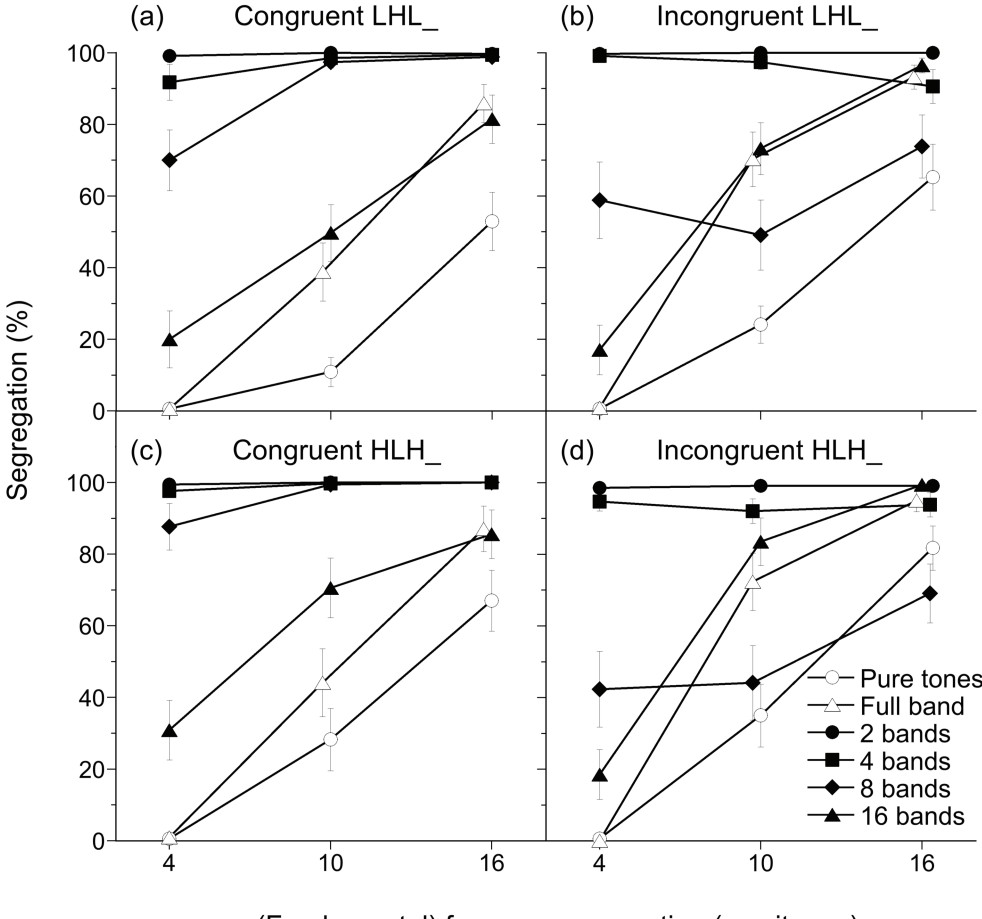

**Fig 7. The results of experiments 1 to 4** ($N$ = 17). The control conditions are irrelevant to congruency. Data points were slightly shifted along the abscissa to show error bars reflecting standard error of the mean (SEM).

These observations were supported by GLMM analysis. A model with random effects of congruency and trial block nested under participant was selected, based on the smallest indices ($-2ResidualLogPseudo-Likelihood$ = 56803.2; generalized $\chi^2/df$ = 1.67). The model revealed fixed effects of (fundamental) frequency separation [$F(2, 8148)$ = 689.64, $p < 0.001$, statistical power = 0.998], control stimulus types [$F(1, 8148)$ = 44.92, $p < 0.001$, statistical power = 0.886], and their interaction [$F(2, 8148)$ = 6.33, $p = 0.002$, statistical power = 0.731]. No other fixed effects reached $p$ levels < 0.05 (S1 Table). Multiple comparisons with Tukey's HSD tests revealed that full-band harmonic complex tones showed greater percentages of segregation responses than pure tones at 10-semitone separation ($t = -18.49$, $p < 0.001$) and 16-semitone separation ($t = -15.21$, $p < 0.001$), but not at 4-semitone separation ($t = 0.6$, $p > 0.999$).

## Stripe-tone conditions

The results of stripe-tone conditions showed the interaction among fundamental frequency separation, number of bands, and congruency. In the congruent conditions [Fig 7(a) and 7(c)], when the number of bands was small, the percentages of segregation responses were constantly high irrespective of fundamental frequency separations. Conversely, when the number of bands became larger, the percentages of segregation responses depended more and more on the fundamental frequency separations. By contrast, the incongruent combinations of the fundamental frequency shifts and the spectral pattern movements [Fig 7(b) and 7(d)] produced a strong interaction effect between fundamental frequency separation and number of bands. The interaction was most obvious between the 8-band stimuli and 16-band stimuli.

Fig 8 clearly shows the effect of congruency on the results. The segregation responses for the 8-band stimuli dropped in the incongruent conditions (especially with 10-semitone separation). In contrast, the segregation responses for the 16-band stimuli increased a bit in the incongruent conditions with 10- and 16-semitone separations. Tone sequence patterns had almost no effect on the results.

These observations were supported by GLMM analysis. A model with random effects of trial block nested under participant and experiment order was selected, based on $-2ResidualLogPseudo-Likelihood$ = 132565.59 and generalized $\chi^2/df$ = 0.67. The model revealed the two-way interaction effect of number of bands × fundamental frequency separation [$F(6, 16272)$ = 9.6, $p < 0.001$, statistical power = 0.874] and the three-way interaction effect of congruency × number of bands × fundamental frequency separation [$F(6, 16272)$ = 22.52, $p < 0.001$, statistical power = 0.926]. No other fixed effects reached $p$ levels < 0.05 (S2 Table). Tukey's HSD tests revealed that congruent 8-band stimuli were more frequently judged to be segregated than incongruent 8-band stimuli at 4-semitone ($t = 11.99$, $p < 0.001$) and 10-semitone separation ($t = 12.51$, $p < 0.001$). By contrast, with 16 bands, incongruent stimuli were more frequently judged to be segregated than congruent stimuli at 10-semitone ($t = -7.88$, $p < 0.001$) and 16-semitone separation ($t = -6.84$, $p < 0.001$). However, congruent 16-band stimuli may be more frequently judged to be segregated at 4-semitone separation ($t = 3.66$, $p = 0.046$). All other comparisons at 4-, 10-, and 16-semitone separations between the corresponding conditions concerning congruency and number of bands resulted in $p$ levels exceeding 0.05. Moreover, incongruent 8-band stimuli were more frequently judged to be segregated than incongruent 16-band stimuli at the 4-semitone separation ($t = 13.78$, $p < 0.001$); however, incongruent 16-band stimuli were more frequently judged to be segregated than incongruent 8-band stimuli at 10-semitone ($t = -13.19$, $p < 0.001$) and 16-semitone separations ($t = -9.02$, $p < 0.001$).

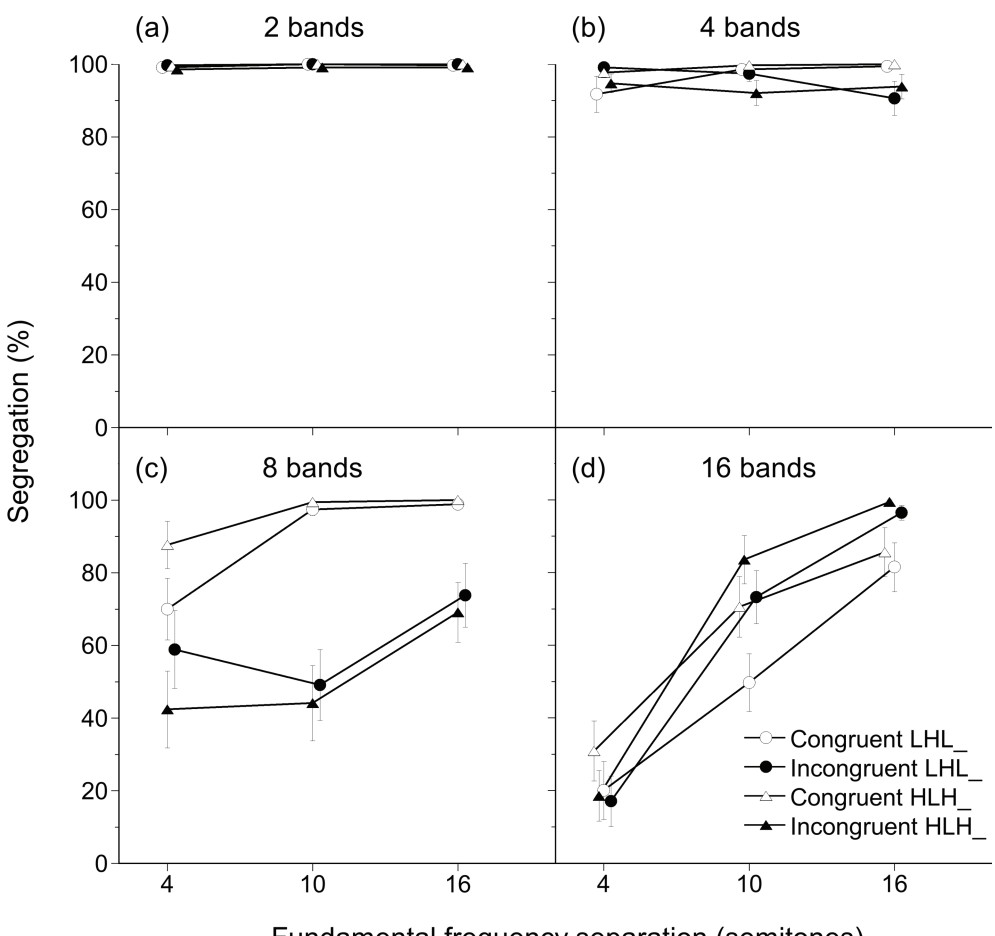

**Fig 8. Replotted percentages of segregation responses of the stripe-tone conditions.** The data in Fig 7 were replotted. (a–d) 2-, 4-, 8-, and 16-band stimuli. Error bars reflect SEM.

## Discussion

### Replications

A larger (fundamental) frequency separation, thus a greater pitch difference, caused more frequent segregation responses for both pure-tone and full-band stimuli. Additionally, full-band stimuli with many harmonic components tended to be more segregated than pure-tone stimuli. Rajasingam et al. [19] and Roberts and Haywood [21] found that two-component complex tones were perceived to be more segregated than pure tones. The current results seem to be in line with those of previous studies.

### Pitch and timbre trade-off in stream segregation

The interaction effect of fundamental frequency separation, number of bands, and congruency was observed for stripe-tone conditions. This means that the pitch and timbre trade-off on sequential stream segregation was observed overall. The most obvious trade-off between pitch and timbre was observed with the eight-band stimuli. These results suggest that the

auditory system captures the spectral patterns switching every 80 ms and groups the same patterns as a stream. However, the auditory system has difficulty combining blocks of components dispersed over the spectrotemporal domain. When the number of bands increased further, the stimuli got closer to full-band stimuli, leading to similar response patterns.

Previous findings on a trade-off between pitch and timbre by shifting four consecutive harmonics [7], shifting a single formant peak [32], and tilting spectral slopes [34] are consistent with the current findings. Moreover, we showed that timbre can be a strong cue for stream segregation against a pitch difference as small as four semitones, which normally leads to an integrated percept of the three-triplet tone sequences [13–16]. Manipulating the number of bands was reflected in the percentages of segregation responses. Thus, the first research question, "Do timbre differences in stripe tones affect the percentages of segregation responses?" was answered with "Yes."

The answer to the second research question, "Does congruency in shift directions of pitch and timbre for stripe tone sequences affect percentages of segregation responses?" is that "It depends on the number of bands." No effect of congruency was observed for two- and four-band stimuli: These stimuli were always perceived to be segregated, irrespective of the sizes of pitch separations and congruency. Listeners perceived congruent eight-band stimuli as segregated most of the time. However, they reported segregation less frequently for incongruent eight-band stimuli. In contrast, their proportions of segregation responses varied mainly according to the sizes of pitch separations for 16-band stimuli: Segregation responses increased as pitch separations became greater. Proportions of segregation responses became greater for incongruent stimuli than for congruent stimuli with 10 or 16 semitone separations, very probably due to the absence of any frequency components in the lowest band of L tones for the congruent 16-band stimuli [Table 2 and Fig 2(h)]. Still, the differences due to congruency were slight. Thus, when the number of bands is two or four, prominent timbre differences between odd- and even-numbered-band stripe tones should mainly govern perceptual segregation. When the number of bands becomes eight or more, timbre differences between odd- and even-numbered-band stripe tones become less competitive with pitch differences, leading to the segregation responses being more governed by pitch differences.

During the revision process of the current paper, we were acknowledged that our another paper on sequential stream segregation by using *band tones* was accepted for publication [38]. Band tones are harmonic complex tones that consist of odd- or even-numbered bands with fixed passbands (comparable to Table 1), unlike the fixed harmonic number groups used for stripe tones (Table 2). With band tones, listeners perceived eight-band tones as segregated most of the time, except for the congruent four-semitone separation. Moreover, with eight bands, segregation responses were reported more often for band tones than for stripe tones. Thus, the current results indicated that the trade-off between pitch and timbre on stream segregation appeared with a sharper contrast with stripe tones than band tones, suggesting that the effects of pitch and timbre differences on perceptual segregation and integration are somewhat balanced around eight-band stimuli.

The answer to the third question, "Do the stripe tone sequence patterns, LHL_ and HLH_, interact with the congruency of pitch and timbre shift directions?" was "No." Tone sequence patterns had practically no effect on segregation, replicating previous results by Thomassen et al. [20] with pure tones.

It is out of the scope of the current investigation whether the participants' responses were based on pitch, timbre, or both because the experimental task was to judge whether they perceived one stream or two streams. Nevertheless, it is worth considering that the current experimental paradigm mimics a situation similar to a two-talker or two-instrument alternation with two pitches. When the number of bands is small, the situation looks somewhat similar

to a two-talker or two-instrument alternation, and the auditory system tends to segregate two streams. Whereas, as the number of bands increases, the situation gets similar to a single-talker or single-instrument alternation, then the auditory system integrates two tones into one stream more frequently, depending on the fundamental frequency separations and thus pitch separations. Obviously, the auditory system shows limits in integrating spectrotemporally scattered harmonics.

## Conclusions

The effects of pitch and timbre separation on sequential stream segregation were investigated. To make the timbral separation, harmonic complex tones with 35 frequency components were divided into 2 to 16 bands based on harmonic numbers, harmonics in every other band were removed, and the resulting two possible stripe-like patterns were alternated with each tone. The stimuli with a few bands elicited strong stream segregation against pitch proximity. By contrast, the results for the stimuli with 16 bands were dominated by pitch separation, similar to full-band control stimuli. The trade-off between pitch and timbre on stream segregation appeared most clearly in the results for eight-band stimuli. The results suggest that the auditory system captures rapidly changing spectral patterns and groups sounds with similar spectral patterns. At the same time, the results also suggest that the auditory system has limits in integrating blocks of frequency components dispersed over frequency and time with a small number (four or fewer) of bands. The current investigation formed a basis for further investigations on detecting an auditory target in a noisy background.

## Supporting information

**S1 Fig. Spectrograms of audio exemplars.** LHL_ triplets with pure tones and full-band harmonic complex tones (a–b and c–d) were used in both experiments 1 and 2. HLH_ triplets with pure tones and full-band harmonic complex tones (e–f and g–h) were used in both experiments 3 and 4. The left column represents exemplars with 2-semitone separation for explaining the one-stream concept, and the right column represents exemplars with 18-semitone separation for explaining the two-stream concept.
(TIF)

**S2 Fig. Spectrograms of control stimuli.** LHL_ triplets with pure tones and full-band harmonic complex tones (a–c and d–f) were used in both experiments 1 and 2. HLH_ triplets with pure tones and full-band harmonic complex tones (g–i and j–l) were used in both experiments 3 and 4. The columns are in the order of 4-, 10-, and 16-semitone separation from left to right.
(TIF)

**S3 Fig. Spectrograms of two-band stripe tones.** LHL_ triplets with congruent and incongruent stripe tones (a–c and d–f) were used in experiments 1 and 2. HLH_ triplets with congruent and incongruent stripe tones (g–i and j–l) were used in experiments 3 and 4.
(TIF)

**S4 Fig. Spectrograms of four-band stripe tones.** LHL_ triplets with congruent and incongruent stripe tones (a–c and d–f) were used in experiments 1 and 2. HLH_ triplets with congruent and incongruent stripe tones (g–i and j–l) were used in experiments 3 and 4.
(TIF)

**S5 Fig. Spectrograms of eight-band stripe tones.** LHL_ triplets with congruent and incongruent stripe tones (a–c and d–f) were used in experiments 1 and 2. HLH_ triplets with congruent and incongruent stripe tones (g–i and j–l) were used in experiments 3 and 4.
(TIF)

**S6 Fig. Spectrograms of 16-band stripe tones.** LHL_ triplets with congruent and incongruent stripe tones (a–c and d–f) were used in experiments 1 and 2. HLH_ triplets with congruent and incongruent stripe tones (g–i and j–l) were used in experiments 3 and 4.
(TIF)

**S1 Data. Data.**
(XLSX)

**S1 Table. GLMM analysis summary for the control conditions.**
(PDF)

**S2 Table. GLMM analysis summary for the stripe-tone conditions.**
(PDF)

## Acknowledgments

The authors would like to thank Yoshitaka Nakajima for providing J language software routines, and Hikaru Eguchi for providing a valuable framework for the LiveCode program.

## Author contributions

**Conceptualization:** Geng-Yan Jhang, Kazuo Ueda.

**Data curation:** Geng-Yan Jhang, Kazuo Ueda.

**Formal analysis:** Geng-Yan Jhang, Kazuo Ueda.

**Funding acquisition:** Geng-Yan Jhang, Kazuo Ueda.

**Investigation:** Geng-Yan Jhang.

**Methodology:** Geng-Yan Jhang, Kazuo Ueda.

**Project administration:** Kazuo Ueda.

**Resources:** Kazuo Ueda.

**Software:** Geng-Yan Jhang.

**Supervision:** Kazuo Ueda.

**Validation:** Geng-Yan Jhang, Kazuo Ueda.

**Visualization:** Geng-Yan Jhang.

**Writing – original draft:** Geng-Yan Jhang.

**Writing – review & editing:** Geng-Yan Jhang, Kazuo Ueda, Hiroshige Takeichi, Gerard B. Remijn, Emi Hasuo.

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
