## [Decision Letter · Decision Letter 0]

11 Nov 2024

PONE-D-24-04194Rivalry between fundamental frequency separation and switching frequency bands for auditory stream segregationPLOS ONE

Dear Dr. Jhang,

Thank you for submitting your manuscript to PLOS ONE. After careful consideration, we feel that it has merit but does not fully meet PLOS ONE’s publication criteria as it currently stands. Therefore, we invite you to submit a revised version of the manuscript that addresses the points raised during the review process.

We look forward to receiving your revised manuscript.

Kind regards,

Si Chen

Academic Editor

PLOS ONE

Journal Requirements:

“This research was supported by the Japan Society for the Promotion of Science (JSPS; https://www.jsps.go.jp) KAKENHI Grant No. JP19H00630 for Kazuo Ueda, and by the Japan-Taiwan Exchange Association (https://www.koryu.or.jp) with a scholarship for Geng-Yan Jhang under the supervision of Kazuo Ueda. The funders had no role in study design, data collection and analysis, decision to publish, or preparation of the manuscript.”

3. We are unable to open your Supporting Information file S1_Fig.eps, S2_Fig.eps, S3_Fig.eps, S4_Fig.eps, S5_Fig.eps and S6_Fig.eps Please kindly revise as necessary and re-upload.

Additional Editor Comments:

Thank you for submitting your manuscript to PLoS One. The manuscript presents an interesting research topic and may potentially contribute to the field. However, several significant issues need to be addressed as pointed out by the two reviewers before it can fully meet the requirement of publication.

The reviewer's comments are attached and please address the comments in a response letter and revise your manuscript accordingly. We look forward to receiving your revised manuscript and response letter. Thank you!

Reviewers' comments:

Reviewer's Responses to Questions

**Comments to the Author**

1. Is the manuscript technically sound, and do the data support the conclusions?

Reviewer #1: Partly

Reviewer #2: Partly

2. Has the statistical analysis been performed appropriately and rigorously? 

Reviewer #1: Yes

Reviewer #2: I Don't Know

3. Have the authors made all data underlying the findings in their manuscript fully available?

Reviewer #1: Yes

Reviewer #2: Yes

4. Is the manuscript presented in an intelligible fashion and written in standard English?

Reviewer #1: No

Reviewer #2: Yes

5. Review Comments to the Author

Reviewer #1: This paper addresses an important topic in the area of acoustics. I really appreciate the careful consideration the authors have given to the experimental design, which is clearly described and executed. To make the paper compelling, there are some areas where further clarification or additional detail could be beneficial, which I have uploaded as an attachment.

Reviewer #2: The manuscript presents an interesting study investigating the interaction between fundamental frequency (F0) separation and switching frequency bands on auditory stream segregation. The design of the four experiments is good, and the experiments were properly executed. However, it could benefit from clearer justifications for some methodological choices, e.g., the specific range of F0 separations and the exact harmonic numbers chosen for stimuli. Also, the number of participants is relatively small, and their experience of musical/listening training is also diverse. This can significantly influence the results of the experiments. My major concern over this paper is the number of participants (17 only). I strongly recommend the authors to increase the number or to present a power analysis of the results. Also, the influence of gender and age has not been fully illustrated. 

The data supports the conclusions, i.e., stream segregation depends more on timbral contrast when fewer frequency bands are used, whereas F0 separation dominates for higher numbers of bands. I would suggest a more detailed discussion on the practical significance of the findings. There is some discussion between Line 413 and 419, which could be further expanded.

A minor comment is that the writing style can be improved by making some sentences more academic and the methodology part more concise.

Line-to-line comments

Line 2-4: You may want to make the first two sentences more academic. Also, without mentioning auditory analysis, “frequency” can be confusing. 

Line 21: “Whereas, if Ls and Hs are perceptually grouped into one stream, do you mean “L-H-L-H-...”?

Line 23: The introduction can be improved by stating the theoretical and practical importance of segregation—why do we need to promote it?

Line 35: You may want to explain pure vs. harmonic complex tones to the naive audience.

Line 39: In fact, it is from this line that I realised the importance of F segregation. You may want to move this up. (See my comment on Line 23.)

Line 95: I am not sure about the statistical power of this sample size, especially since their music-training and listening-training backgrounds are complex. Also, any gender difference?

Line 101: Absolute pitch, in my understanding, is quite important to the results of the present studies. A short test rather than self-reporting data will improve the validity of the results. 

Line 128: How were the exemplars introduced to the audience?

Line 326: “Participants might have misunderstood...” Is this misunderstanding purposefully? Is it that they could not help but base their responses on pitch perception? If it is the latter, you may want to discuss the auditory/cognitive indication of this mistake.

Line 375: “Thus, the combinations of these three variables affected sequential stream segregation.” This sentence is important but could be illustrated more. What do you mean by "combinations," and what are the real-world implications?

Line 396-397: “A transition from a segregated percept to an integrated percept was realised by increasing the number of frequency bands for the stimuli” sounds like an improvement. Is it an improvement? If not, please be careful with the tone of description in the manuscript.

6. PLOS authors have the option to publish the peer review history of their article (what does this mean?). If published, this will include your full peer review and any attached files.

Reviewer #1: No

Reviewer #2: No

---

## [Author Response · Author response to Decision Letter 1]

28 Feb 2025

Thank you very much for the constructive comments on the manuscript. Our responses were submitted as a response letter in a PDF file.

---

## [Decision Letter · Decision Letter 1]

17 Apr 2025

Rivalry between pitch and timbre in auditory stream segregation

PONE-D-24-04194R1

Dear Dr. Jhang,

We’re pleased to inform you that your manuscript has been judged scientifically suitable for publication and will be formally accepted for publication once it meets all outstanding technical requirements.

Kind regards,

Si Chen

Academic Editor

PLOS ONE

Additional Editor Comments (optional):

Reviewers' comments:

Reviewer's Responses to Questions

**Comments to the Author**

1. If the authors have adequately addressed your comments raised in a previous round of review and you feel that this manuscript is now acceptable for publication, you may indicate that here to bypass the “Comments to the Author” section, enter your conflict of interest statement in the “Confidential to Editor” section, and submit your "Accept" recommendation.

Reviewer #1: All comments have been addressed

Reviewer #2: All comments have been addressed

2. Is the manuscript technically sound, and do the data support the conclusions?

Reviewer #1: Yes

Reviewer #2: Yes

3. Has the statistical analysis been performed appropriately and rigorously? 

Reviewer #1: Yes

Reviewer #2: Yes

4. Have the authors made all data underlying the findings in their manuscript fully available?

Reviewer #1: Yes

Reviewer #2: Yes

5. Is the manuscript presented in an intelligible fashion and written in standard English?

Reviewer #1: Yes

Reviewer #2: Yes

6. Review Comments to the Author

Reviewer #1: The authors have made great efforts to address the issues I raised, which is much appreciated. Glad to see the manuscript has improved significantly.

Reviewer #2: While musical training was modeled as a nested random effect, it might still be interesting to analyze it as a fixed effect in future studies (e.g., via median split into “low” and “high” training). Given that the task relies on perceptual grouping—often enhanced by musical experience—this could reveal subtle interactions - you may want to add this into limitation/future study.

The authors mention that their findings may have implications for speech-in-noise perception, but this point could be expanded

However, these are just minor suggestions that the authors may consider to revise before publicaiton.

7. PLOS authors have the option to publish the peer review history of their article (what does this mean?). If published, this will include your full peer review and any attached files.

Reviewer #1: No

Reviewer #2: No

---

## [Editor Report · Acceptance letter]

PONE-D-24-04194R1

PLOS ONE

Dear Dr. Jhang,

I'm pleased to inform you that your manuscript has been deemed suitable for publication in PLOS ONE. Congratulations! Your manuscript is now being handed over to our production team.

Kind regards,

on behalf of

Dr. Si Chen

Academic Editor

PLOS ONE